# Differential Effects of Overexpression of Wild Type and Kinase-Dead MELK in Fibroblasts and Keratinocytes, Potential Implications for Skin Wound Healing and Cancer

**DOI:** 10.3390/ijms24098089

**Published:** 2023-04-30

**Authors:** Łukasz Szymański, Krystyna Lieto, Robert Zdanowski, Sławomir Lewicki, Jean-Pierre Tassan, Jacek Z. Kubiak

**Affiliations:** 1Department of Molecular Biology, Institute of Genetics and Animal Biotechnology, Polish Academy of Sciences, Postępu 36A, 05-552 Magdalenka, Poland; 2Laboratory of Molecular Oncology and Innovative Therapies, Department of Oncology, Military Institute of Medicine—National Research Institute, 04-141 Warsaw, Poland; 3Institute of Outcomes Research, Maria Sklodowska—Curie Medical Academy, 03-411 Warsaw, Poland; 4Faculty of Medical Sciences and Health Sciences, Kazimierz Pulaski University of Technology and Humanities in Radom, 26-600 Radom, Poland; 5Dynamics and Mechanics of Epithelia Group, Institute of Genetics and Development of Rennes (IGDR), CNRS, University Rennes, UMR 6290, 35043 Rennes, France

**Keywords:** MELK, ERK, p38, MAPK, JNK, p53, AKT1, MAPK9, keratinocytes, fibroblasts, proliferation, cell cycle, wound healing

## Abstract

Maternal embryonic leucine-zipper kinase (MELK) plays a significant role in cell cycle progression, mitosis, cell migration, cell renewal, gene expression, embryogenesis, proliferation, apoptosis, and spliceosome assembly. In addition, MELK is known to be overexpressed in multiple types of cancer and is associated with cancer proliferation. Tumorigenesis shares many similarities with wound healing, in which the rate of cell proliferation is a critical factor. Therefore, this study aimed to determine the involvement of MELK in the regulation of cell division in two cell types involved in this process, namely fibroblasts and keratinocytes. We examined how temporal overexpression of wild-type and kinase-dead MELK kinase variants affect the rate of proliferation, viability, cell cycle, and phosphorylation state of other kinases involved in these processes, such as ERK1/2, AKT1, MAPK9, p38, and p53. We explored if MELK could be used as a therapeutic stimulator of accelerated wound healing via increased proliferation. We observed that aberrant expression of MELK results in abnormal proliferation, altered cell cycle distribution, and decreased viability of the cells, which challenge the utility of MELK in accelerated wound healing. Our results indicate that, at least in healthy cells, any deviation from precisely controlled MELK expression is harmful to fibroblasts and keratinocytes.

## 1. Introduction

Maternal embryonic leucine-zipper kinase (MELK) is a conserved, cell cycle-dependent protein kinase that plays a crucial role in many cellular processes such as cell cycle progression, mitosis, cell migration and renewal, embryogenesis, proliferation, and spliceosome assembly. Its exact role in cell physiology is still largely unknown, but in addition to regulating the cell cycle, cell division, and proliferation, it is involved in apoptosis and gene expressions such as ERK 1 and 2 [1]. Thus, MELK is a multifunctional kinase that is also involved in tumorigenesis induction due to its multidirectional interaction with different types of proteins in the cell. MELK is known to be overexpressed in multiple types of cancer, including breast cancer, melanoma, and renal cell carcinoma [2]. At the protein level, MELK is hyperphosphorylated and reaches maximal activity during the M phase of the cell cycle in human cancer lines [3] and embryonic cells [4,5]. Moreover, Ye et al. showed, using esophageal squamous cell carcinoma cell line KYSE-150, that wild-type cells closed the wound after 16 h, while MELK deficient cells were unable to close the wound in the wound healing assay, thus demonstrating that MELK is required for cell migration [6]. Tumorigenesis shares many similarities with other processes during which loss of cell polarity, extensive tissue remodeling, and increased proliferation are of key importance, such as wound healing [7]. Thus, many molecular and cellular pathways are common between wound healing and tumorigenesis. Wound healing is a highly complex yet perfectly coordinated process that enables the restoration of physical tissue integrity after injury. Two major skin cell types, fibroblasts, and keratinocytes, play crucial yet different roles in wound healing. Fibroblasts participate in critical processes such as extracellular matrix formation, fibrin clot breakdown, wound contraction, and collagen structure formation. On the other hand, keratinocytes are responsible for re-epithelialization and are involved in the complex mechanisms of initiation, maintenance, and completion of wound healing. The role of each cell type and the importance of crosstalk between these cells was described in great detail by Amiri et al. [8] and Rodrigues et al. [9]. Cell proliferation is one of the key steps in tissue formation during wound healing. During the proliferation phase, the reepithelization and neovascularization of the wound occur [10]. The number of “new” cells that are needed is usually considerable; however, the proliferation rate is physiologically controlled, which distinguishes wound healing from tumorigenesis, where the intensified proliferation is chaotic. Knowing that, at least in some cancers, MELK expression is associated with increased proliferation, we hypothesized that an increase in MELK level may lead to a higher rate of proliferation and consequently accelerate wound healing. Therefore, the purpose of this study was to determine the involvement of the serine/threonine protein kinase MELK in the regulation of cell division in normal fibroblasts and keratinocytes, the key actors of wound healing, by examining how temporal changes in cellular MELK kinase (active vs. kinase dead) levels affect the rate of proliferation, and whether MELK could be used as a potential therapeutic stimulator of accelerated wound healing.

## 2. Results

In this study, we overexpressed wild-type (WT) and kinase-dead (KD) human MELK tagged with GFP in normal human fibroblasts and keratinocytes. Throughout the whole study, only GFP-positive cells were analyzed. The GFP expression, indicating MELK WT’s and MELK KD’s overexpression, persisted from 6 h to 72 h after transfection. The study scheme is presented in Figure 1.

Figures presenting a combined side-by-side comparison of fibroblasts and keratinocytes are available in Appendix A as Appendix A.

### 2.1. Fibroblasts

#### 2.1.1. Effects of MELK Overexpression on BJ-5ta Cell Viability

The viability of BJ-5ta fibroblast cells was significantly lower both in MELK WT and KD transfected cells at all time points as compared to control cells transfected with GFP alone. Overexpression of MELK WT lowered the viability of the cells to a greater extent than overexpression of MELK KD at 6 h, 24 h, 48 h, and 72 h. Interestingly, MELK WT overexpression caused high levels of apoptosis (Annexin V^+^/PI^−^ and Annexin V^+^/PI^+^) 6 h after transfection, followed by a predominantly necrotic (Annexin V^−^/PI^+^) type of death at 24 h, 48 h, and 72 h, while in MELK KD overexpressed cells, the type of cell death was almost exclusively necrotic. Results are presented in Figure 2.

#### 2.1.2. Effects of MELK Overexpression on BJ-5ta Cell Cycle Phase Distribution and Proliferation Rate

Overexpression of MELK WT resulted in a decrease in the percentage of BJ-5ta cells in G1 6 h, 24 h, 48 h, and 72 h after transfection, with a simultaneous increase of the percentage of G2 cells at 24 h, 48 h, and 72 h. In MELK KD, the percentage of G1 cells was reduced 24 h and 48 h post-transfection, with an increase in G2 cells only after 24 h from transfection. Interestingly, overexpression of MELK WT led to an increased percentage of cells in S phase 6 h and 72 h after transfection, while MELK KD did not. In addition, no significant changes in the proliferation rate were observed in MELK WT overexpressing cells, while overexpression of MELK KD resulted in a significant increase in proliferation rate 24 h after transfection. The results are presented in Figure 3.

#### 2.1.3. Effects of MELK Overexpression on Selected Signaling Pathways Activation

Overexpression of MELK WT resulted in increased relative phosphorylation of AKT1 48 h and 72 h, MAPK9 (JNK) 24 h, 48, 72 h, and p38 MAPK 24 h after transfection. On the other hand, MELK KD overexpression leads to increased phosphorylation of MAPK9 (JNK) 24 h and 72 h, p38 MAPK 24 h and p53 24 h and 48 h after transfection. No significant changes were observed in phosphorylation of ERK1/2 after WT and KD MELK overexpression or in AKT1 phosphorylation after overexpression of MELK KD. Results are presented in Figure 4.

#### 2.1.4. Intracellular Localization of MELK WT and MELK KD in BJ-5ta Cells

We found MELK WT, as detected using GFP fluorescence, to be either concentrated near the nucleus or uniformly distributed in the cytoplasm of BJ-5ta cells. MELK KD, on the other hand, was observed solely in the condensed form near the nucleus. Representative pictures are presented in Figure 5.

### 2.2. Keratinocytes

#### 2.2.1. Effects of MELK Overexpression on Ker-CT Cell Viability

Compared to the control, the viability of MELK WT keratinocyte KerCT cells was significantly lower 6 h, 48 h, and 72 h after transfection, while the overexpression of MELK KD caused a decrease in Ker-CT cells’ viability only after 48 h and 72 h. At 48 h and 72 h, the decrease in viability was significantly higher in MELK WT cells compared to MELK KD cells. The cell death was primarily caused by apoptosis and occurred a long time after transfection, suggesting its link with prolonged overexpression. The results are presented in Figure 6.

#### 2.2.2. Effects of MELK Overexpression on Ker-CT Cell Cycle Phase Distribution and Proliferation Rate

Overexpression of MELK WT resulted in a decrease in the percentage of KerCT cells in G1 phase 6 h, 24 h, 48 h, and 72 h and a significant increase in the percentage of G2 cells at 24 h, 48 h, and 72 h after transfection. The changes were more pronounced as time passed after transfection (x% G1: x% G2, 61.52%:10.73%, 67.14%:25.50%, 33.24%:43.52%, 11.17%:68.74%; 6 h, 24 h, 48 h, and 72 h, respectively). In addition, changes in cell cycle phase distribution in MELK WT cells were associated with a significant decrease in proliferation rate 48 h and 72 h post-transfection.

Similarly, overexpression of MELK KD also leads to an increased percentage of cells in the G2 phase, but only at 6 h and 48 h. At 72 h after transfection, a decrease in the percentage of cells in the S phase was observed that was associated with a decrease in proliferation rate. Interestingly, an increase in proliferation rate in MELK KD cells was seen 6 h after transfection, i.e., in a short time following overexpression. Results are presented in Figure 7.

#### 2.2.3. Effects of MELK Overexpression on Selected Signaling Pathways Activation

Overexpression of MELK WT resulted in increased relative phosphorylation of AKT1 24 h, 48 h, 72 h, and p38 MAPK 48 h after transfection. On the other hand, MELK KD overexpression leads to decreased phosphorylation of MAPK9 (JNK) 24 h and increased phosphorylation of p38 MAPK 24 h and 48 h after transfection. No significant changes were observed in the phosphorylation of ERK1/2 and p53 after WT and KD MELK overexpression or in AKT1 phosphorylation after overexpression of MELK KD. Results are presented in Figure 8.

#### 2.2.4. Intracellular Localization of MELK WT and MELK KD in Ker-CT Cells

Similar to fibroblasts, in the Ker-CT cells, we found MELK WT to be either concentrated near the nucleus or in the cytoplasm, where it was uniformly distributed. On the other hand, the expression of MELK KD was observed solely in the condensed form near the nucleus. Representative pictures are presented in Figure 9.

## 3. Discussion

MELK, also known as MPK38 or pEg3 in Xenopus, is involved in various cell functions, including cell cycle, apoptosis, and proliferation. Its role in cell division and daughter cell fate in Xenopus levis is well known [11]. Interestingly, MELK has also emerged as a potential target in cancer research as its expression is significantly elevated in many different tumors, such as glioblastoma [12], melanoma [13], breast [14], prostate [15], and renal cancer [16]. MELK expression level correlates with malignancy grade, and MELK is a marker of cancer stem cells and is generally associated with poor prognosis [6,17,18,19]. On the other hand, in a CRISPR/Cas9 screen, deletion of MELK did not affect the cancer cells, calling into question its essentiality in cancer induction and cancer cells survival [20,21]. The controversies regarding MELK’s role in cancer were partially settled, at least in triple-negative breast cancer cells, by Wang et al., who showed that MELK is essential for clonogenic proliferation, whereas under standard culture conditions, its silencing has negligible effects. Interestingly, the same pattern of cell influence was observed in mutated KRAS and MYC, which are well-known oncogenes [22]. While MELK’s role in embryogenesis and cancer was extensively studied, no thought was given to its role in normal skin cells and the possibility of using this kinase as a therapeutic stimulator for accelerated wound healing. To the best of our knowledge, the current study is the first investigation of the role of MELK and the effects of MELK overexpression in skin cells. Moreover, as most of the studies of MELK were carried out in cancer cells, embryonic cells, and following genetic knockout in a mouse organism, our study sheds new light on the role of MELK in healthy, non-transformed somatic cells in cell culture. It is of importance for the comparison with cancer cells cultured in vitro, which do not have the same requirements and different properties.

Overexpression of MELK WT induced an increase in the percentage of cells in G2/M and concomitant reduction in the G1 phase, consistent with a G2/M arrest and/or prolongation. These results agree with previously published work by Davezac et al. and Gray et al. [23,24] on cancer cells. MELK KD led to an initial increase in proliferation in both keratinocytes and fibroblasts; however, these effects were transient. Of note, proliferation analysis through the Ki67 staining allows for a snapshot of the analyzed cell’s state but does not consider the change in the number of cells over time. Specifically, to calculate the relative proliferation rate, we measured Ki67 expression, which is upregulated in dividing cells; however, this signal is also present in cells that were ready to divide but, for some reason, such as cytokinesis problem or cell cycle arrest, did not. Therefore, the proliferation results should be evaluated together with viability data. It seems that the spike in the relative proliferation index caused by the overexpression of MELK KD in fibroblast was associated with the increase in p53 phosphorylation [25]. The MELK KD overexpression may stimulate fibroblast proliferation, but the concomitant p53 phosphorylation seems to enhance apoptosis and the cell cycle arrest in an effort to maintain homeostasis of the state before the overexpression [26]. This result suggests a dialogue between MELK and p53 in the regulation of the level of apoptosis and cell cycle progression. The fact that MELK WT overexpression may escape the p53 control contrary to the KD MELK overexpression may suggest that the kinase activity of MELK plays an important role in this potential feedback loop. It may also point to the fact that the buildup of not fully functional MELK protein is unfavorable to the cells, hence the observed activation of p53, a stress-sensing protein. The elevated relative proliferation rate, measured as the expression of Ki67 protein, is consistent with observed changes in cell cycle phase distribution, namely a decrease in G1 and an increase in S and G2 [27]. Our results suggest that the presence of higher MELK levels is a signal for a cell to proliferate; however, it is the excessive kinase activity that results in robust G2/M arrest. Our fluorescence analysis results in which MELK WT, but not MELK KD, relocates from the cytoplasm to the cell’s cortex or vice versa seem consistent with its proposed roles in G2/M transition and cytokinesis regulation [11,28]. The ability of MELK WT to relocate to the cytoplasm, which MELK KD does not possess, points to the presence of a short amino acid sequence we termed the “Mitotic Localization Signal” (MLS), which regulates MELK localization (unpublished results). Through fluorescence microscopy, we showed that MLS-deficient MELK KD is unable to localize to the cytoplasm of keratinocytes and fibroblasts. Our results also indicate that the presence of the MLS in the MELK structure is responsible for the abnormal cell cycle phase distribution induced by MELK WT overexpression. Interestingly, Davezac et al. observed that both overexpression and knockdown of WT MELK in U2OS cells lead to a similar G2/M delay and suggested that MELK might be required for G2/M checkpoint progression, but its presence later becomes inhibitory for proliferation [23]. This agrees with our observations that aberrant MELK expression leads to decreased proliferation, polyploidy, and altered cell cycle phase distribution. Finally, we observed significant MELK kinase activity-dependent phosphorylation of AKT1 at serine 473 that was associated with G2/M arrest. It is known that AKT1 is needed for G2/M transition, but some reports indicate that AKT1 also induces DNA replication without mitosis [29,30]. Therefore, whether the increase in AKT1 phosphorylation during MELK WT-induced G2/M arrest is a cell’s mechanism of overcoming the G2/M arrest or is a mechanism of the arrest itself needs further clarification.

Overexpression of MELK WT and, to a lesser extent, of MELK KD led to a significant reduction in the viability of both keratinocytes and fibroblasts. Interestingly, in fibroblasts, overexpression of MELK led to a predominantly necrotic type of death, while in keratinocytes, the type of death was primarily apoptotic. Over the years, p38 MAPK and MAPK9 (JNK) were shown to activate or inhibit apoptosis [31,32]. However, the ratio of p38 to JNK was also shown to affect cells’ proliferation and differentiation [33]. In our study, in Ker-CT cells, we observed a short-term decrease in MAPK9 phosphorylation and a long-term increase in p38 phosphorylation associated with apoptosis of the transfected cells. The results are consistent with the findings of Jung et al., who showed that overexpression of MELK results in JNK and p38-mediated increase in H_2_O_2_ and leads to apoptosis [34]. On the other hand, in BJ-5ta cells, we observed a long-term increase in the phosphorylation of MAPK9 and a short-term increase in the phosphorylation of p38 related to the necrosis of the cells. Production of reactive oxygen species and altered expression and phosphorylation of MAPK9/JNK and p38 was shown to induce necrosis under various circumstances [35,36,37]. In particular, Sakon et al. showed that the accumulation of reactive oxygen species with concomitant inhibition of the NF-κB pathway favors necrosis in fibroblasts [38]. What is more, MAPK9/JNK pathways have been shown to play a role in fibroblast migration and keratinocyte migration, differentiation, and proliferation [39]. p38 has also been shown to regulate the migration of the cells during wound healing [40,41,42]. Unfortunately, the current literature is conflicting with respect to the role of MAPK9/JNK and MAPK p38 signaling in fibroblast and keratinocyte behavior, and further work is required. Therefore, we propose that the duration of activation and the ratio of p38 to MAPK9 may determine cell fate, including the type of death. It is also possible that cell-type-specific differences arise from the physiological predispositions of the cells. Keratinocytes, which, under certain circumstances, form syncytium, are characterized by greater plasticity than fibroblasts [43]. Due to this plasticity, keratinocytes may be more resistant to G2/M arrest, cytokinesis failure, and polyploidy, leading to “silent” apoptosis, while in less resistant fibroblasts, it leads to “louder” necrosis. Upon stress induction caused by DNA damage, change in redox potential, osmotic or heat shock MAPK9/JNK is known to phosphorylate its substrates, including p53, which leads to cell death [44]. Presented results showing that phosphorylation of MAPK9/JNK associated with phosphorylation of p53 occurs in fibroblasts but not keratinocytes further strengthen this hypothesis. Cell-type-specific influence of MELK overexpression on cell fate does not support the possibility of using MELK as a common factor stimulating the proliferation process in the skin.

In this research, we showed that overexpression of neither MELK WT nor MELK KD is a viable concept for accelerated wound healing. Nevertheless, we observed that aberrant expression of MELK results in abnormal proliferation, altered cell cycle phase distribution, and decreased viability of cells. Our observations indicate that, at least in healthy cells, the deviation from precisely controlled MELK expression is harmful to the cells. Therefore, these preliminary results might suggest that MELK is not an independent driver of proliferation and, thus, oncogenesis, which further explains the failure of MELK inhibitors in some clinical trials [21]. Further studies are needed to confirm this hypothesis.

### Limitations

Although two-dimensional, monolayer cultures of immortalized cells allow for a broad and reproducible research spectrum, such culture conditions do not mimic the physiological features of the tissues and cell-to-cell and cell-to-ECM interactions, which are crucial during wound healing. In the present study, we used (hTERT)-immortalized primary cells that combine the in vivo nature of primary cells with the ability to survive continuously in an in vitro culture without causing cancer-associated changes or altering phenotypic properties [45]. Nevertheless, these culture conditions create an artificial system; therefore, further studies should be performed on more advanced 3D tissue models containing keratinocytes and fibroblasts. Moreover, this study used transient transfection of MELK WT and MELK KD, resulting in a mixed population of transfected and untransfected cells limiting the ability to perform functional, cell-to-cell interaction, and cell behavior studies. The results presented here represent preliminary findings in decoding the complicated role of MELK in skin cells and wound healing. Therefore, further studies investigating the function of MELK in wound healing should be performed on stably transfected (overexpression and knockdown/knockout) cells. Since the siRNA targeting MELK is known to have high off-target effects, the use of CRISPR to knock out the gene is preferable [21]. During our current work to decipher the role of MELK in wound healing, we designed 3 AAV-DJ vectors, each targeting a different MELK exone to be used simultaneously to achieve a complete knockout. All of the 1500 clonally cultured Ker-CT keratinocytes died. Interestingly, the problem with clonal expansion was less pronounced in BJ-5ta fibroblasts. The analysis of a few surviving BJ-5ta clones revealed that all were partial knockouts with one copy of unchanged MELK remaining. Proliferation dependency on MELK under clonogenic conditions has been shown for MDA-MB-468 cells [22,46]. Since the MELK clonogenic dependency was observed in non-transformed somatic cells as well as in cancer cells, this phenomenon is worth studying in detail due to the putative functional biological dependency of cell proliferation on cell–cell interaction. Finally, in further studies, the focus should be given to MELK activation status and localization during wound healing as well as to cell-to-cell interactions and migration abilities.

## 4. Materials and Methods

Reagents were purchased from ThermoFisher Scientific, Warsaw, Poland unless stated otherwise. A detailed list of reagents is presented in the description of each method. The gating strategy for cytometric analysis is available in the Appendix A in Appendix A.

### 4.1. Cell Culture

BJ-5ta and KerCT, hTERT-immortalized cell lines used in this research were acquired from the ATCC collection. Cells were cultured as described previously [47]. Briefly, cells were cultured under standard aseptic conditions (37 °C, 95% humidity, 5% CO_2_) using KGM-Gold medium with supplements and penicillin/streptomycin for KerCT cells and a 1:4 mixture of Medium 199 and DMEM with the addition of 10% FBS, penicillin/streptomycin, and 0.01 mg/mL hygromycin B for BJ-5ta cells. After retrieving the cells from cryostorage, the cells were passaged at least twice before experiments. EVE Plus Automated Cell Counter (NanoEntek, Seoul, Korea) was used for cell counting.

### 4.2. Expression Constructs

Wild-type (WT) and kinase-dead (KD) MELK constructs were manufactured by Biomatic, Ontario, Canada, by cloning MELK cDNA (GeneBank accession number: BC014039), using BamHI–AgeI cloning sites into the pEGFP-N1 plasmid. KD MELK construct was prepared by introducing K40R mutation (aaa into cgc). MELK K40R KD mutation was previously described [34]. Plasmids were replicated using competent DH5α strain and standard heat shock procedure [48]. The empty pEGFP-N1 plasmid was used as the negative control.

### 4.3. Transfection

BJ-5ta cells were transfected using Neon Transfection System set to 1650 V, 20 ms pulse width, and 1 pulse. KerCT cells were transfected with plasmids using Viromer RED (Lipocalyx GmbH, Halle, Germany) according to the manufacturer’s recommendations.

### 4.4. Cell Viability

Cell viability was measured according to the procedure described elsewhere [47]. Briefly, after trypsinization, the cells were washed and resuspended in 50 μL of PBS containing 1 μg/sample of propidium iodide (PI, BD Bioscience, Warsaw, Poland) and 4.5 μM/sample APC-conjugated annexin V (BD Bioscience, Warsaw, Poland). Next, cells were incubated at 4 °C and analyzed using flow cytometry (CytoFlex, Beckman Culter, Warsaw, Poland). A minimum of 10,000 GFP-positive cells was analyzed. Finally, the results were analyzed using Kaluza (version 2.1. Beckman Coulter, Warsaw, Poland) and presented as the mean SD of at least three independent experiments (*n* ≥ 9).

### 4.5. Proliferation (Ki67 Assay) and Cell Cycle

After trypsinization, the cells were washed twice with PBS. The cells were fixed for 15 min using ice-cold 0.5% PFA, washed with PBS, permeabilized in 200 µL of cold 70% methanol, and stored in a −20 °C freezer before analysis. On the day of analysis, the samples were washed with PBS containing 0.5% BSA. Ki67 PI staining was performed by adding 50 µL of PBS containing 0.5% BSA and 5 µL of Ki67-APC antibody to each sample and incubated for 30 min RT in the dark. Next, the samples were washed, and 50 µL of the FxCycle PI/RNase Staining Solution was added for 30 min. A minimum of 10,000 GFP-positive cells were analyzed for each sample. The results were analyzed using Kaluza (Beckman Coulter, Poland) and presented as the mean SD of at least three independent experiments (*n* ≥ 9).

### 4.6. Cell Signaling—Phosphorylation

As described above, after trypsinization, the cells were washed, fixed, permeabilized, and stored before analysis. On the day of analysis, samples were washed with PBS containing 0.5% BSA. Stainings were performed by adding 100 µL of PBS containing 0.5% BSA and 1 µL of appropriate antibody to each sample and incubated for 60 min RT in the dark. Following antibodies were used: Phospho-AKT1 (Ser473) REF: 17-9715-42, Phospho-p53 (Ser15) REF: MA5-36930, Phospho-ERK1/2 (Thr202, Tyr204) REF: 17-9109-42, Phospho-p38 MAPK (Thr180, Tyr182) REF: 17-9078-42, Phospho-MAPK9 (Thr183, Tyr185) REF: MA5-37135. Cells were analyzed by flow cytometry (CytoFlex, Beckman Culter, Warsaw, Poland). A minimum of 10,000 GFP-positive cells was analyzed. Finally, results were analyzed using Kaluza (version 2.1. Beckman Coulter, Warsaw, Poland).

### 4.7. Cell Imaging

Transfected BJ-5ta cells were cultured in 8-well Nunc Chamber Slides for 24 h, 48 h, and 72 h. The cells were fixed for 15 min using ice-cold 0.5% PFA, washed with PBS, permeabilized in 200 µL of cold 70% methanol, and stained with ActinRed 555 according to the manufacturer’s recommendation. The slides were mounted with SlowFade Antifade with DAPI.

Fluorescence photographs of BJ-5ta were made using Leica SP8 confocal microscope and LAS X software ver. 3.7.1 (magnification 630×), and Ker-CT fluorescence photographs were made using Nikon A1R confocal microscope and NIS-Elements ver. 5.30.00 software (magnification 600×).

### 4.8. Statistical Analysis

All results were presented as the mean ± standard deviation (SD). The data distribution was evaluated using the Shapiro–Wilk test. Statistical evaluation of viability and the proliferation rate was performed using the Kruskal–Wallis test with Dunn’s multiple comparison test. Cell cycle results were analyzed using two-way ANOVA with Tukey’s multiple comparisons test. Finally, protein phosphorylation data were evaluated using two-way ANOVA with Dunnett’s multiple comparisons tests. GraphPad Prism software (version 9.4.0; GraphPad Software, Inc., La Jolla, CA, USA) was used for all evaluations. *p* < 0.05 was considered statistically significant.

## Figures and Tables

**Figure 1 ijms-24-08089-f001:**
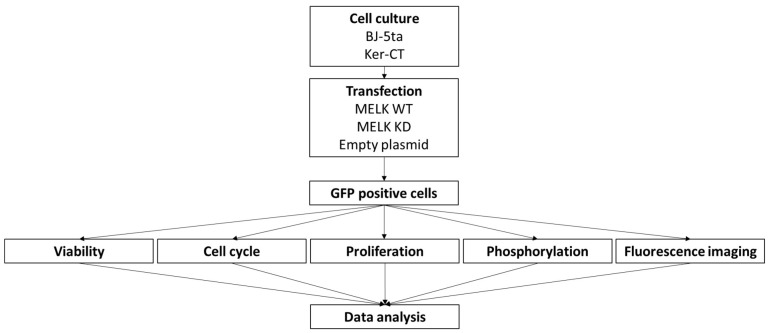
Schematic representation of study design.

**Figure 2 ijms-24-08089-f002:**
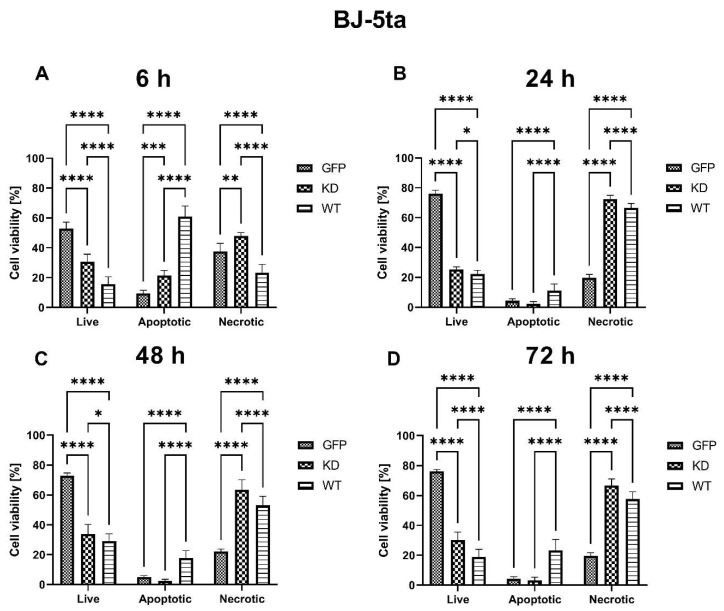
Effects of MELK WT, MELK KD, and empty plasmid overexpression on cell viability (percentage of live, apoptotic, and necrotic cells) in BJ-5ta fibroblasts. (**A**) 6 h after transfection. (**B**) 24 h after transfection. (**C**) 48 h after transfection. (**D**) 72 h after transfection. Results are presented as the mean SD of at least three independent experiments (*n* ≥ 9). * *p* ≤ 0.05, ** *p* ≤ 0.01, *** *p* ≤ 0.001, **** *p* ≤ 0.0001.

**Figure 3 ijms-24-08089-f003:**
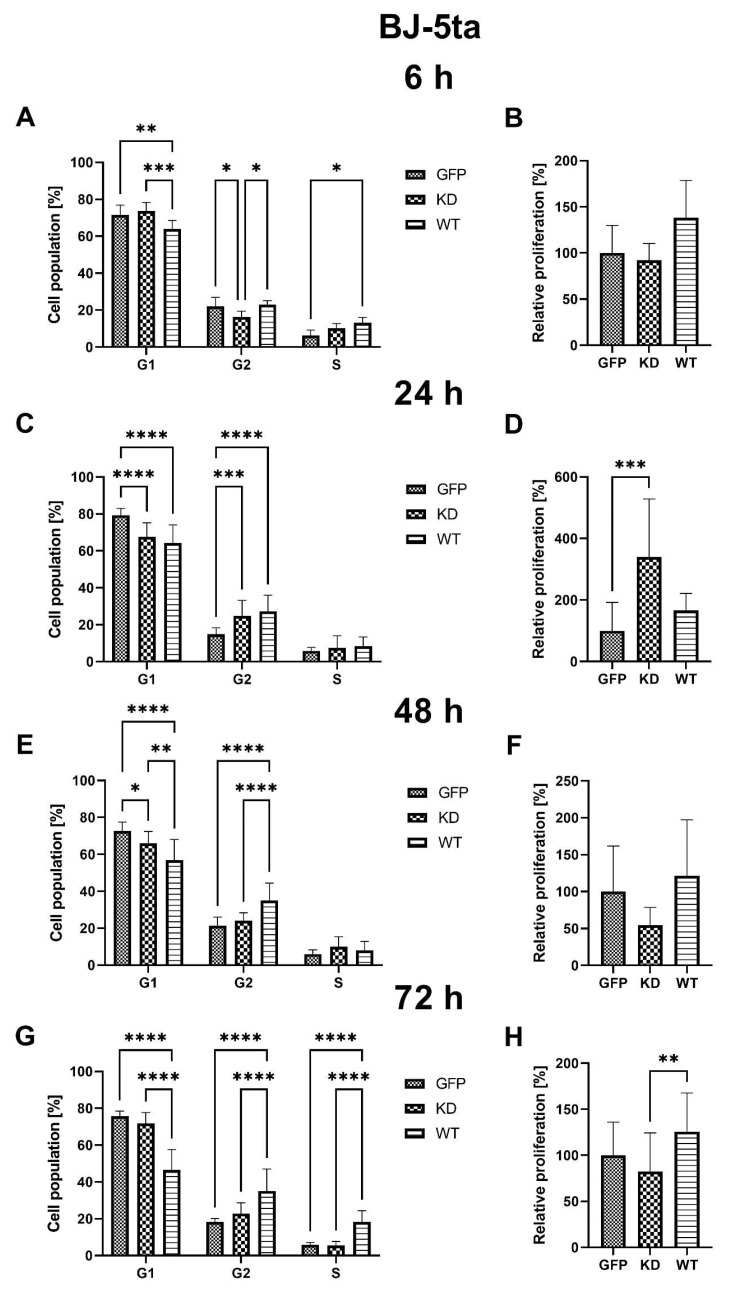
Effects of MELK WT, MELK KD, and empty vector overexpression on cell cycle distribution (percentage of cells in G1, G2 and S phase) and proliferation rate in BJ-5ta fibroblasts. (**A**) cell cycle analysis 6 h after transfection. (**B**) relative proliferation 6 h after transfection. (**C**) cell cycle analysis 24 h after transfection. (**D**) relative proliferation 24 h after transfection. (**E**) cell cycle analysis 48 h after transfection. (**F**) relative proliferation 48 h after transfection. (**G**) cell cycle analysis 72 h after transfection. (**H**) relative proliferation 48 h after transfection. Results are presented as the mean SD of at least three independent experiments (*n* ≥ 9). * *p* ≤ 0.05, ** *p* ≤ 0.01, *** *p* ≤ 0.001, **** *p* ≤ 0.0001.

**Figure 4 ijms-24-08089-f004:**
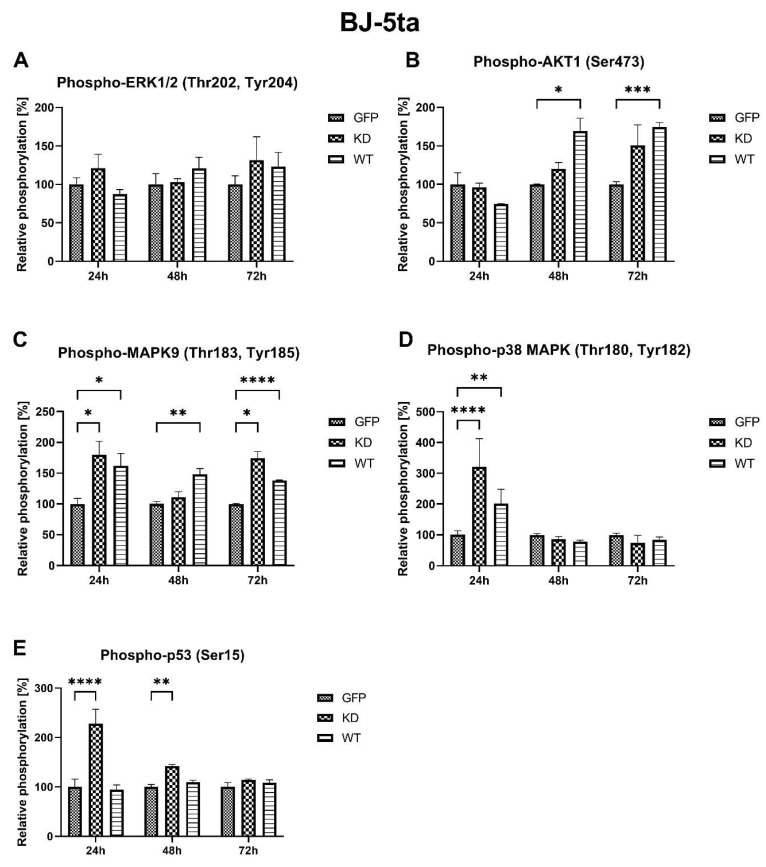
Effects of MELK WT and MELK KD overexpression on relative, compared to empty vector-transfected cells, phosphorylation of ERK1/2, AKT1, MAPK9 (JNK), p38 MAPK, and p53 in BJ-5ta fibroblasts. (**A**) ERK1/2 (Thr202, Tyr204) in cells 24 h, 48, and 72 h after transfection. (**B**) (Ser473) in cells 24 h, 48, and 72 h after transfection. (**C**) MAPK9/JNK (Thr183, TYR185) in cells 24 h, 48, and 72 h after transfection. (**D**) MAPK (Thr180, Tyr182) in cells 24 h, 48, and 72 h after transfection. (**E**) p53 (Ser15) in cells 24 h, 48, and 72 h after transfection. Results are presented as the mean SD of at least three independent experiments (*n* = 6). * *p* ≤ 0.05, ** *p* ≤ 0.01, *** *p* ≤ 0.001, **** *p* ≤ 0.0001.

**Figure 5 ijms-24-08089-f005:**
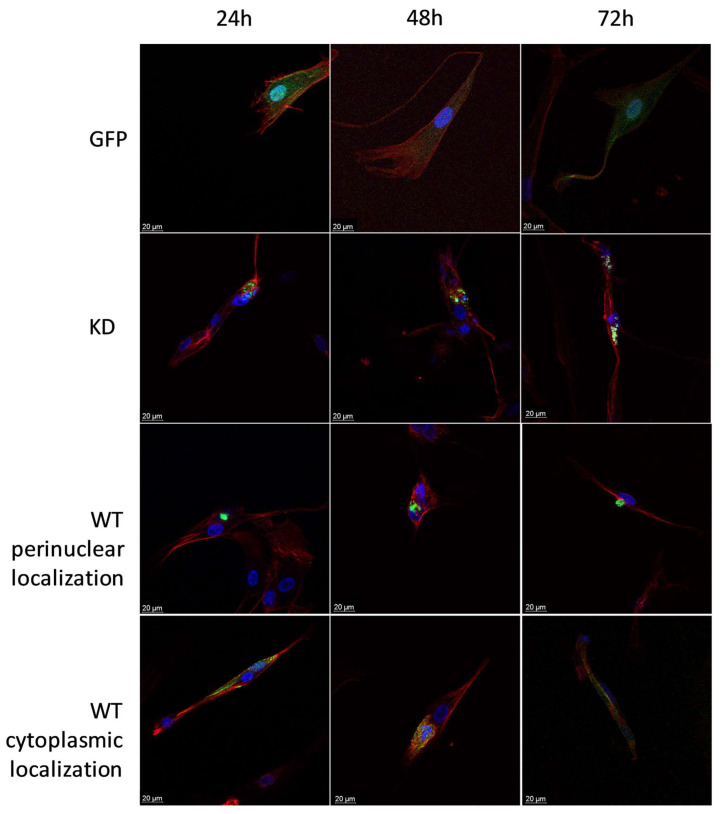
MELK WT and MELK KD localization in BJ-5ta cells 24, 48, and 72 h after transfection. Green–GFP fluorescence; Blue–nucleus stained with DAPI; Red–F-actin stained with rhodamine-phalloidin.

**Figure 6 ijms-24-08089-f006:**
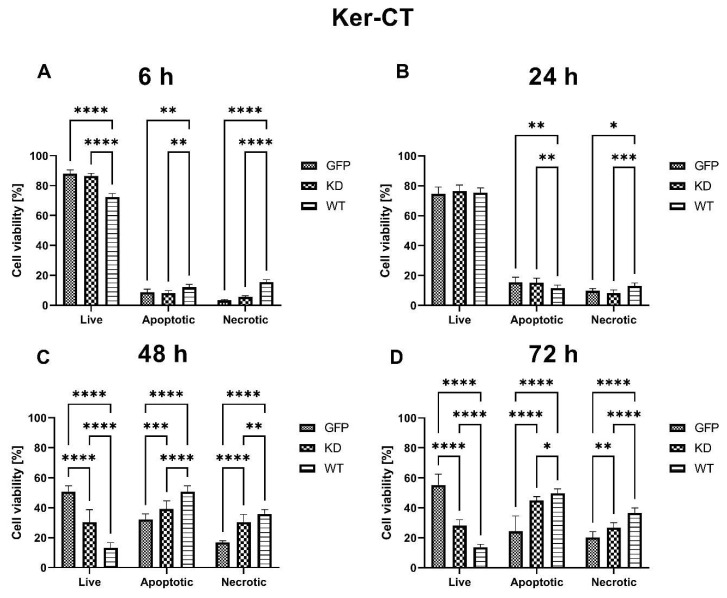
Effects of MELK WT, MELK KD, and empty plasmid overexpression on cell viability (percentage of live, apoptotic, and necrotic cells) in Ker-CT keratinocytes. (**A**) 6 h after transfection. (**B**) 24 h after transfection. (**C**) 48 h after transfection. (**D**) 72 h after transfection. Results are presented as the mean SD of at least three independent experiments (*n* ≥ 9). * *p* ≤ 0.05, ** *p* ≤ 0.01, *** *p* ≤ 0.001, **** *p* ≤ 0.0001.

**Figure 7 ijms-24-08089-f007:**
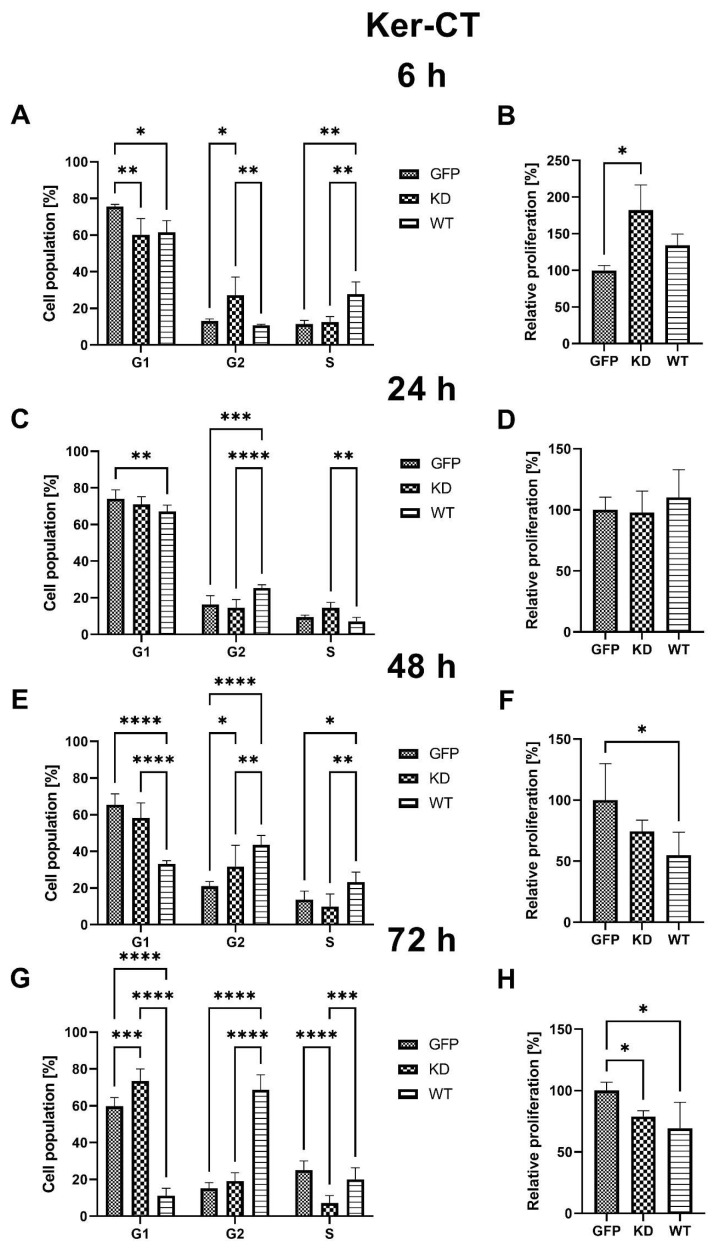
Effects of MELK WT, MELK KD, and empty vector overexpression on cell cycle distribution (percentage of cells in G1, G2, and S phase) and proliferation rate in Ker-CT keratinocytes. (**A**) Cell cycle analysis 6 h after transfection. (**B**) Relative proliferation 6 h after transfection. (**C**) Cell cycle analysis 24 h after transfection. (**D**) Relative proliferation 24 h after transfection. (**E**) Cell cycle analysis 48 h after transfection. (**F**) Relative proliferation 48 h after transfection. (**G**) Cell cycle analysis 72 h after transfection. (**H**) Relative proliferation 48 h after transfection. Results are presented as the mean SD of at least three independent experiments (*n* ≥ 9). * *p* ≤ 0.05, ** *p* ≤ 0.01, *** *p* ≤ 0.001, **** *p* ≤ 0.0001.

**Figure 8 ijms-24-08089-f008:**
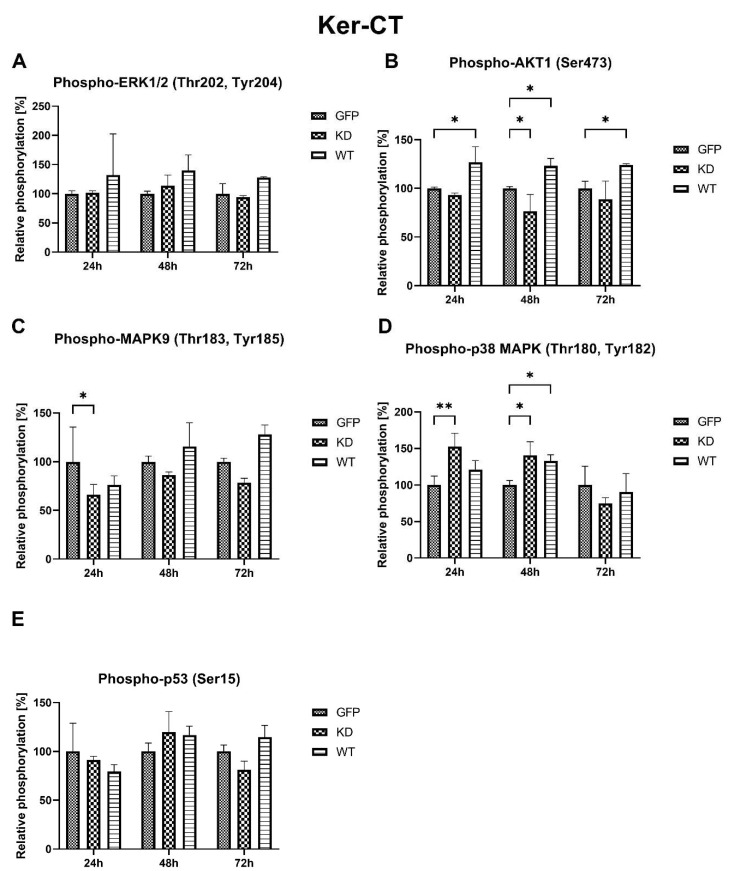
Effects of MELK WT and MELK KD overexpression on relative, compared to empty vector transfected cells, phosphorylation of ERK1/2, AKT1, MAPK9 (JNK), p38 MAPK, and p53 in Ker-CT keratinocytes. (**A**) ERK1/2 (Thr202, Tyr204) in cells 24 h, 48, and 72 h after transfection. (**B**) (Ser473) in cells 24 h, 48, and 72 h after transfection. (**C**) MAPK9/JNK (Thr183, TYR185) in cells 24 h, 48, and 72 h after transfection. (**D**) MAPK (Thr180, Tyr182) in cells 24 h, 48, and 72 h after transfection. (**E**) p53 (Ser15) in cells 24 h, 48, and 72 h after transfection. Results are presented as the mean SD of at least three independent experiments (*n* = 6). * *p* ≤ 0.05, ** *p* ≤ 0.01.

**Figure 9 ijms-24-08089-f009:**
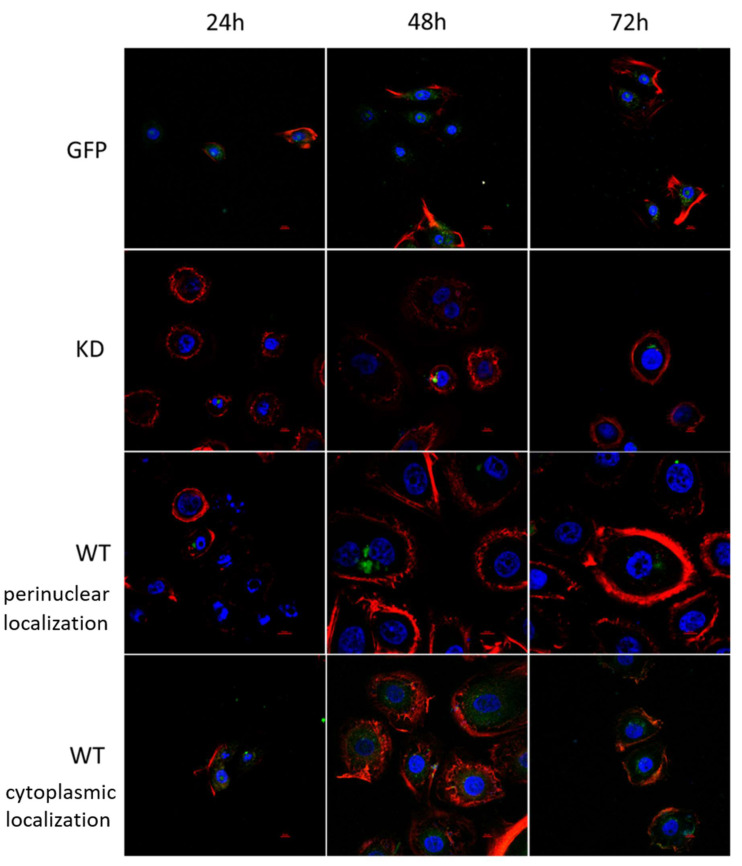
MELK WT and MELK KD localization in Ker-CT cells 24, 48, and 72 h after transfection. Green–GFP fluorescence; Blue–nucleus stained with DAPI; Red–F-actin stained with rhodamine-phalloidin.

## Data Availability

The datasets generated during and/or analyzed during the current study are available from the corresponding author upon reasonable request.

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
