# Peer review of "Differential Effects of Overexpression of Wild Type and Kinase-Dead MELK in Fibroblasts and Keratinocytes, Potential Implications for Skin Wound Healing and Cancer"

_ijms, 2023, doi:10.3390/ijms24098089_

Round 1

Reviewer 1 Report

In this paper, the authors conducted a functional analysis of MELK using normal keratinocytes and fibroblasts, suggesting that optimal activation of MELK is necessary to exert its appropriate effects on these cells. This reviewer thinks this study contributes to elucidating the control mechanism of MELK functions for researchers in this field and is worthy of publication in the International Journal of Molecular Sciences.

However, the quantitative analysis of MELK wild-type and KD mutants on their intracellular localization, in Figure 5, is necessary before publication.

Author Response

Response to the Reviews

We want to thank the Reviewers for all their valuable comments that allowed us to improve our manuscript. We tried to incorporate as many improvements in the manuscript as possible. All changes of earlier text are marked in tracking changes and are included in the current manuscript text according to all reviewer suggestions. Below, we also include all reviewers' comments with our responses.

First round

Reviewer 1

In this paper, the authors conducted a functional analysis of MELK using normal keratinocytes and fibroblasts, suggesting that optimal activation of MELK is necessary to exert its appropriate effects on these cells. This Reviewer thinks this study contributes to elucidating the control mechanism of MELK functions for researchers in this field and is worthy of publication in the International Journal of Molecular Sciences.

  1. However, the quantitative analysis of MELK wild-type and KD mutants on their intracellular localization, in Figure 5, is necessary before publication.

Response: Thank you for the suggestion regarding the localization of MELK WT and KD. We also believe that the topic is interesting however, in the present research we focused on quantitative evaluation using flow cytometry. Using confocal microscopy, we wanted to highlight our observations of MELK localization. In the future, we plan to investigate it further using continuous live cell imaging and high content analyzer (which was unfortunately not included in the grant proposal financing this research) of MELK WT, MELK KD, and native MELK, which will allow for quantitative analysis of MELK localization in different cellular compartments. Currently, we suspect that the localization of MELK in skin cells is cell cycle dependant and is associated with cytokinesis as in Xenopus embryo, which we described here https://www.ncbi.nlm.nih.gov/pmc/articles/PMC3798187/. Chartrain et al., Cell-cycle dependent localization of MELK and its new partner RACK1 in epithelial versus mesenchyme-like cells in Xenopus embryo. Biol Open. 2013 Oct 15; 2(10): 1037–1048. Based on our data, we believe that there is a short amino acid sequence we call MLS, which regulates MELK localization. The cell membrane localization was confirmed both in Xenopus levis normal cells XL2 and in human cancer cells HeLa, but it was not shown in non-transformed human cells (https://www.ncbi.nlm.nih.gov/pmc/articles/PMC3798187/ Chartrain, Couturier and Tassan. Cell-cycle-dependent cortical localization of pEg3 protein kinase in Xenopus and human cells. Biol Cell. 2006 Apr;98(4):253-63; under the previous name of MELK - pEg3). After acquiring new funding, we plan to investigate the localization of MELK during different cell cycle stages using both immunofluorescence and biochemical assays in non-transformed human cells because this kind of research may explain intriguing differences between the role of MELK in cells upon physiological and pathological conditions.

Reviewer 2 Report

1. Abstract. Add '(MELK)' in line 17.

2. Figure 2. The transfection seems very toxic. Only 50% of control (GFP) cells are viable (6 h). Is it correct? The toxicity of transfection itself should take into consideration for data analysis.

3. Representative plots for flow cytometry should be provided. This provides helpful information such as how the threshold of positive is determined.

4. Discuss why/how kinase-dead MELK affects cell cycle/viability.

5. Discuss why WT/KD MELK decreases viability (figure 2) but does not decrease proliferation (figure 3).

6. It is necessary to knock down MELK or use a small molecule inhibitor to validate the results. Will knockdown or inhibit MELK show the opposite result of MELK overexpression?

Round 2

Reviewer 1 Report

This reviewer agrees with the authors' opinion that further analysis is needed to determine the cellular localization of MELK. However, the image of BJ-5ta cells in the first column and third row of the panel in Figure 5 is complicated for this reviewer to understand, as multiple cells overlap and the signal around the nucleus of MELK-WT needs to be clarified. It is strongly recommended to replace it with a picture with a single cell as other pictures in the panel.

The typos pointed out in the first round of review have not been corrected.
Line 367,
"in a (-20°C)" should be "in a -20°C freezer."
Lines 116 and 181,
"r relative" should be "relative." 

Author Response

Response to the Reviews

We once again want to thank the Reviewers for all their valuable and rapid evaluation of resubmitted manuscript. We tried to incorporate as many improvements in the manuscript as possible. All changes of earlier text are marked in tracking changes and are included in the current manuscript text according to all reviewer suggestions. Below, we also include all reviewers' comments with our responses.

Second round

Reviewer 1

  1. This reviewer agrees with the authors' opinion that further analysis is needed to determine the cellular localization of MELK. However, the image of BJ-5ta cells in the first column and third row of the panel in Figure 5 is complicated for this reviewer to understand, as multiple cells overlap and the signal around the nucleus of MELK-WT needs to be clarified. It is strongly recommended to replace it with a picture with a single cell as other pictures in the panel.

Response: We have replaced the image of 24h MELK WT transfected fibroblasts with perinuclear localization of the protein with a new one showing a single cell more clearly, as requested by the Reviewer.

  1. The typos pointed out in the first round of review have not been corrected.
    Line 367,
    "in a (-20°C)" should be "in a -20°C freezer."
    Lines 116 and 181,
    "r relative" should be "relative." 

Response: We have corrected that.

Reviewer 2 Report

I appreciate the responses. However, I only see a few grammatical changes in the discussion. If the responses will be published together with the paper, please ignore this comment. If not, I would like to see the responses incorporated into the paper. 

Author Response

Response to the Reviews

We once again want to thank the Reviewers for all their valuable and rapid evaluation of the resubmitted manuscript. We tried to incorporate as many improvements in the manuscript as possible. All changes of earlier text are marked in tracking changes and are included in the current manuscript text according to all reviewer suggestions. Below, we also include all reviewers' comments with our responses.

Second round

Reviewer 2

  1. I appreciate the responses. However, I only see a few grammatical changes in the discussion. If the responses will be published together with the paper, please ignore this comment. If not, I would like to see the responses incorporated into the paper.

Response: We have incorporated the responses into the text however, we will also request for the review reports to be published with the manuscript.